# BLAB: BRUTALLY LONG AUDIO BENCH

## ABSTRACT

Developing large audio language models (LMs) capable of understanding diverse spoken interactions is essential for accommodating the multimodal nature of human communication and can increase the accessibility of language technologies across different user populations. Recent work on audio LMs has primarily evaluated their performance on short audio segments, typically under 30 seconds, with limited exploration of long-form conversational speech segments that more closely reflect natural user interactions with these models. To address this gap, we introduce Brutally Long Audio Bench (BLAB), a challenging long-form audio reasoning benchmark that evaluates audio LMs on localization, duration estimation, emotion and counting tasks using audio segments averaging 51 minutes in length. BLAB consists of 833+ hours of diverse, full-length audio clips, each paired with human-annotated, text-based natural language questions and answers. Our audio data were collected from permissively licensed sources and underwent a human-assisted filtering process to ensure task compliance. We evaluate six open-source and proprietary audio LMs on BLAB, and find that all of them, including advanced models such as Gemini 2.0 Pro and GPT-4o, struggle with the tasks in BLAB. Our comprehensive analysis reveals key insights into the trade-offs between task difficulty and audio duration. In general, we find that audio LMs struggle with long-form speech, with performance declining as duration increases. They perform poorly on localization, temporal reasoning, speaker counting, and struggle to understand non-phonemic information, relying more on prompts than audio content. BLAB serves as a challenging evaluation framework to develop audio LMs with robust long-form audio understanding and reasoning capabilities.

## 1 INTRODUCTION

Comprehensive audio perception and reasoning are central to building intelligent agents capable of real-world interaction. Text and visual inputs alone cannot fully capture the richness of human communication, which is multimodal and requires reasoning over a wide range of auditory cues, including tone, pitch, and rhythm. Recent audio language models (LMs; Chu et al., 2024; Gemini Team et al., 2024; OpenAI et al., 2024; Ghosh et al., 2025; Microsoft et al., 2025) have demonstrated strong audio comprehension skills on short audio clips up to 30 seconds (Huang et al., 2024; Sakshi et al., 2024) or at most 5 minutes (Ghosh et al., 2025). This leaves a critical gap: real-world audio, such as meetings, music, podcasts, and other media content, routinely spans tens of minutes to hours. Segmenting long recordings into short chunks may appear practical, but it limits the ability to benchmark reasoning over broader contexts, produces fragmented or inconsistent predictions, and introduces substantial computational and post-processing overhead. While long-context modeling has been extensively studied for text (Dong et al., 2024) and visual inputs (Chen et al., 2025), in the audio domain, long-form analyses are limited to conversational speech recognition (Cornell et al., 2025b), underscoring the need for systematic evaluation of long-form audio reasoning.

To address this gap, we introduce Brutally Long Audio Bench (BLAB), the first benchmark dedicated to reasoning over long-form audio (15 minutes–2 hours). BLAB contains over 833 hours of conversational speech across eight tasks and evaluates four fundamental reasoning skills: temporal localization, speaker counting, emotion interpretation, and duration estimation (see Figure 1 for an overview). Our task selection was guided by practical real world scenarios in which audio LMs could be expected to perform end-to-end reasoning over long-form content. We focused on tasks that require not just recognition of words or sounds, but understanding temporal structure, contextual relationships, and

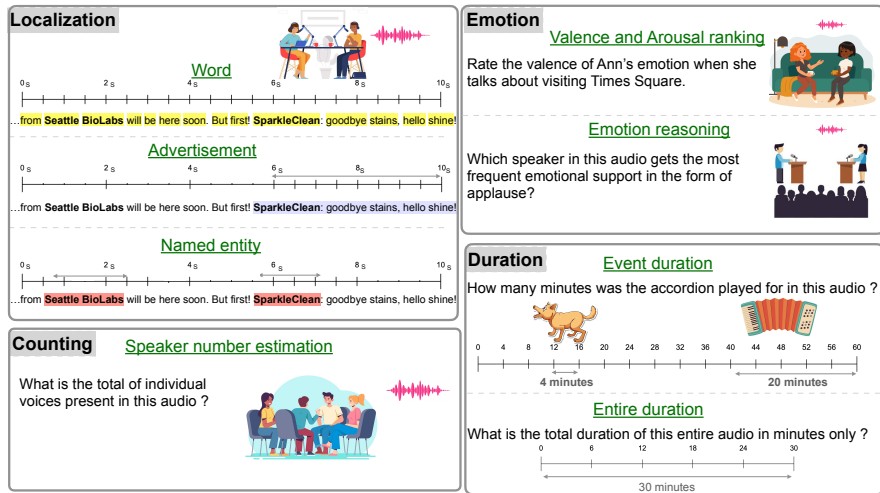

Figure 1: Overview of BLAB, designed to test true long-context multimodal understanding abilities of audio LMs. It contains eight distinct audio tasks across four categories, namely **localization**, **counting**, **emotion**, and **duration estimation**. [†]

acoustic cues across extended audio spans. These tasks have high impact applications. For instance, media platforms like YouTube create timestamped chapters and highlights to help users find relevant content quickly, and robust temporal localization abilities could make this process even more effective for long-form audio. Compared to existing benchmarks, BLAB is substantially more challenging due to the extensive length of the audio samples, which contain richer contextual information. Our data is entirely sourced from Creative Commons-licensed videos on YouTube, using a rigorous human-assisted filtering procedure to ensure diverse and high-quality content (more details in §2).

Using BLAB, we conduct a comprehensive evaluation (§4) and in-depth analysis (§5) of several frontier audio LMs. Our analysis reveals that even proprietary models achieve an average $F_1$ score up to **3.02** on localization tasks (Gemini 2.0 Flash) and average exact match accuracy up to **22.25** on the remaining tasks (Gemini 2.0 Pro), underscoring the complexity of our benchmark and the limitations of current modeling approaches. We thoroughly analyze model responses across all tasks and document key patterns, common errors, and areas where models struggle the most. We find that audio duration plays a large role in model performance in BLAB, as well as task complexity. Even though all models struggle to perform tasks in BLAB, we still observe considerable performance gaps between open-sourced and proprietary models, especially Gemini. These findings motivate new research on long-form audio. However, limited transparency in most models' training data and checkpoints makes it hard to probe their results. This underscores the need for open-source long-context multimodal LMs with fully documented data, checkpoints, and training methods.

## 2 BLAB: BRUTALLY LONG AUDIO BENCH

The primary focus of BLAB is to evaluate perception and reasoning abilities of audio LMs on long-form audio derived from various real-world sources. Audio-grounded reasoning requires a model to identify and execute skills relevant for solving an audio task implicitly or explicitly. This motivates us to design eight tasks that demand extended reasoning over long-form audio. One may ask why these tasks cannot be addressed with short audio clips; however, long audio contain contextual dependencies and relationships that are difficult to capture through simple chunking or mere speech recognition. Existing long-form datasets such as CHiME (Barker et al., 2018; Watanabe et al., 2020), AMI (Carletta et al., 2005), and TED (Hernandez et al., 2018) focus on speech recognition. In contrast, BLAB shifts the focus to audio-grounded reasoning. To capture challenging scenarios that require reasoning, we collect audio with diverse properties, including multiple speakers with distinct profiles spread across the recording, overlapping sound events, occasional background music, etc.

Each task includes 200 Creative Commons-licensed audio files sourced from YouTube, alongside corresponding human-annotated questions and answers. Each audio was carefully and thoughtfully selected, prioritizing complexity, quality, diversity, and task relevance. The question-answer pairs were either manually generated by the authors of this paper or model-generated and verified by the authors to ensure quality. We provide a detailed description of each category and its tasks below.

## 2.1 LOCALIZATION TASKS

The localization tasks require identifying the start and end timestamps of specific events within audio samples. They enable real-world applications such as audio indexing, retrieval, targeted advertising, and creating timestamped highlights for videos. allowing users to quickly locate moments in lectures or podcasts without relying on transcripts. In this category, we create three tasks, namely **word localization**, **advertisement localization**, and **named entity (NE) localization**. These tasks have received limited attention in long-form audio and are often restricted to very short audio clips (e.g., 30 seconds) (Huang et al., 2024; Fiscus et al., 1970). We describe our annotation procedure below.

**Word Localization**  We used 200 audio files obtained from YouTube (total duration of 191 hours, samples are 57 minutes on average) and applied existing forced alignment (FA) tools to obtain word- and sentence-level alignments between audio samples and their corresponding transcripts. For word localization, we used WhisperX to generate word-level timestamps for each word spoken in each audio sample (Bain et al., 2023). Next, an annotator (one of the authors) manually reviewed a subset of our entire dataset to ensure that the forced alignments were error-free (only ∼1% of timestamps needed to be corrected). Each audio sample contains 10,500 word-timestamp pairs on average.

**Named Entity Localization**  We defined nine entity categories to be localized: Event, Location, NORP (nationalities or religious or political groups), Organization, Person, Product, TV shows, Temporal, and Work of Art, as well as "All entities", which includes all of the above. For each category, we also define fine-grained subcategories, allowing us to evaluate whether a model understands the nuances of entity types within the same category. For instance, the example in Table 3 focuses on movie entities, which are a subcategory of Work of Art. Next, we crawled 200 audio files from YouTube and transcribed using WhisperX. Each transcript was fed into a text-only LM to extract plausible NE spans for all entity categories. We tested this part of the annotation process with GPT-4 (OpenAI et al., 2024) and Claude-3.5 Sonnet (Cla) and obtained a higher recall with Claude so we settled on Claude. After extracting the NE spans from text, we mapped these spans back to their timestamps (their location in the audio), also derived from WhisperX. We paired 49 audio files from our pool to the "All entities" category. For the single entity categories, we excluded audio files with fewer than 15 predicted entity spans or a duration of less than 20 minutes, leaving 69 audio files from the remaining 151 in our pool. These 69 audios were then paired to multiple "single entity" categories, resulting in 151 extra items for the NE localization task. This yields audio data with a total duration of 110 hours, each sample being 56 minutes on average. Compared to existing work by Huang et al. (2024), which reports an average of 2 entities per sample, our dataset contains an average of 46 entities per sample. The number of entities to be localized and their temporal position in the audio adds to the complexity of this task, as models often struggle to capture long-term dependencies.

**Advertisement Localization**  We used 200 podcasts from YouTube. Similar to the NE localization annotation procedure, we transcribed each podcast using WhisperX. We fed the transcripts into GPT-4 to extract plausible advertisement segments. Since we obtained very long transcripts, we fed them to the model in chunks of 20 sentences at a time, instructing the model to extract segments that contained an advertisement of a product or promotion from the podcast host. We observed that GPT-4 sometimes incorrectly identified segments of the transcript as advertisements, while in other cases it missed spans that should have been detected. To remedy this, an annotator (one of the authors) reviewed all predicted advertisement segments for every audio sample, removed false positives and added segments that were missing. The percentage of advertisement segments that were kept was 84%. Subsequently, we aligned human-verified advertisement segments with their temporal location in the audio file using the sentence-level timestamps from WhisperX. Our final set of audio data consists of 232 hours of audio across 200 podcasts, with an average of three advertisement segments per podcast and 180 words per advertisement segment.

## 2.2 COUNTING TASK

**Speaker Number Estimation**    The task of this category is to count the total number of distinct voices in an audio sample. Existing work cover clips with fewer than 10 speakers (Huang et al., 2024; Sakshi et al., 2024; Yang et al., 2024; Cornell et al., 2025a), but BLAB includes samples with 4–80 speakers, averaging 53 minutes in length. Long audio introduce challenges such as overlapping speech and new speakers appearing later, testing a model's ability to track multiple speakers over extended periods. This task is crucial for applications like speaker diarization, meeting summarization. We included 200 YouTube audio samples (total duration of 177 hours with an average audio sample length of 53 minutes) in this task. Two annotators (also authors) listened to the entire audio samples and counted the number of distinct speakers. Due to the task complexity, some samples received different counts from the annotators. For these cases, a third annotator reviewed both annotations and selected a final count. In majority cases, annotators argued that multiple counts could be valid, so we retained a range of count labels and adjusted our evaluation metric (see Section B.2) to consider any number within the range as correct. Overall, audio samples with a range of values as their ground truth count labels account for 60% of our dataset, and the range does not exceed two speakers.

## 2.3 EMOTION TASKS

The emotion tasks involve ranking emotions expressed in speech and non-verbal sounds, and reasoning over emotional expressions. They evaluate a model's ability to integrate semantic content with acoustic features that are strong indicators of emotions, and absent in text alone. They are critical for real-world applications such as analyzing customer service calls, monitoring sentiment in meetings, enhancing voice assistants' empathetic responses, and detecting emotional cues in podcasts or media.

**Valence & Arousal Ranking**    Existing benchmarks typically structure emotion tasks around discrete emotion classification, using labels such as happy, sad, or angry (Huang et al., 2024; Sakshi et al., 2024; Yang et al., 2024). However, this design does not account for variations in emotional intensity, which is particularly relevant in speech-based emotional expressions (Martínez et al., 2014; Sethu et al., 2019). This motivates us to evaluate audio LMs' ability to rank ordinal emotional expressions in audio segments rather than to classify them. Yannakakis et al. (2017; 2021) presented strong evidence supporting the ordinal nature of emotions, demonstrating that ordinal labels enhance the validity, reliability, and overall performance of emotion recognition models in affective computing. Ordinal emotion labels from classical emotion theory (Russell, 1980; Lotfian & Busso, 2019) are based on ranking emotions by intensity, and are often analyzed along three dimensions: valence (the degree of negativity or positivity in the emotion), arousal (level of activation or intensity), and dominance (control or power over the emotion) with respect to another speaker in a conversation.

To curate our dataset, we used 28 audio samples obtained from YouTube (total duration is 18 hours, and samples are 39 minutes on average) and performed annotation through human-AI collaboration. To increase the complexity of the task, we focused on ranking speech segments in an audio file that are difficult to distinguish in terms of emotional content based off as transcript. This challenges cascaded systems and encourages end-to-end audio LMs to jointly consider the prosodic and semantic content in the audio. Each audio sample in our dataset is diarized into speaker-specific segments using Pyannote (Plaquet & Bredin, 2023; Bredin, 2023), obtaining 600 segments on average per audio sample. To create instances for which labels cannot be inferred from text alone, we input each audio segment into an emotion recognition model trained to predict valence and arousal scores (Wagner et al., 2023). Next, we feed corresponding transcripts to GPT-4 to obtain text-based valence and arousal scores. We filter segments where difference between the audio and text-based valence and arousal scores for the same segment was greater than a threshold (0.3 in our experiments). Two annotators reviewed each segment, verifying its alignment with the expressed emotion. For each sample, we randomly sample up to four segments for evaluating both valence and arousal. We crafted questions for each segment, prompting the model to rank the expressed emotion. This resulted in 156 high-quality segments with verified valence and arousal scores across 18 audio samples.

**Emotional Reasoning**    Our motivation for this task is to evaluate an audio LM's ability to understand emotions beyond surface-level sentiments in long audio, focusing on deeper emotional states and pragmatic meanings of speech over extended periods. We manually identified 22 audio samples on YouTube that were suitable candidates for this task, such as those obtained from controversial debate

podcasts, comedy shows, movie review podcasts, and emotionally charged interviews. One annotator listened to each audio sample and identified emotional patterns and shifts, such as transitions from calmness to agitation, or from happiness to frustration, and crafted reasoning questions based on verbal and non-phonemic cues, like changes in speech tempo, pauses, or arousal. We design this task as a multiple-choice question answering problem, where confounders are generated by extracting plausible, contextually relevant answers from the audio, increasing the task's difficulty. Overall we have 44 questions paired with 22 audio samples and plan to scale further in future work.

### 2.4 DURATION TASKS

**Event Duration**    This task involves predicting the duration of specific acoustic events (e.g., laughter in a comedy special, question-and-answer segments in a panel session, or a particular speaker's total speaking time in a meeting) within an audio sample, or the total duration of the sample itself in seconds. This task evaluates basic temporal reasoning abilities of audio LMs, including their capacity to identify, localize, and track both verbal and non-verbal acoustic events and activities. We used 87 YouTube audio samples (total duration of 77 hours with an average duration of 53 minutes). An annotator (one of the authors) reviewed each audio sample, detecting and labeling acoustic events and activities. The diverse range of genres in YouTube allowed us to identify multiple events within a single audio sample. After selecting these events, the annotator formulated questions and answers and added their specific durations, pairing 200 questions to all 87 audio files. Each audio sample was paired with at least one question and up to a maximum of four questions.

**Entire Duration**    This task asks an audio LM to predict the total duration of an audio file in seconds. The dataset includes 200 YouTube samples ranging from 8 seconds to 92 minutes (92 hours total, 27 minutes on average), making it the only task with short audio. The total duration of audio samples is 92 hours, and samples are 27 minutes on average. While estimating total duration may seem like a stress test, it is key to temporal reasoning. Accurate event duration estimation relies on understanding the overall temporal structure to correctly associate acoustic events with their time frames.

## 3 EXPERIMENTAL SETUP

### 3.1 MODELS

Almost all open-weight audio LMs (Chu et al., 2024; Ghosh et al., 2024b) can only process audio samples with a duration of 30 seconds or less. This limitation comes from their underlying training setup (Radford et al., 2022), which truncates audio samples longer than 30 seconds. Audio Flamingo (Ghosh et al., 2025) is an exception, as it supports audio samples with a duration of up to 5 minutes. In contrast, Gemini models (Gemini Team et al., 2024) support up to 9.5 hours of audio, while GPT-4o (OpenAI et al., 2024) handles up to 8 minutes of audio. Based on these model characteristics and the design of BLAB, our evaluations focus on four Gemini models: Gemini 2.0 Flash, Gemini 2.0 Pro. To enable broader comparisons and analyze the impact of audio duration on model performance, we also evaluate other models (Chu et al., 2024; OpenAI et al., 2024; Ghosh et al., 2025; Microsoft et al., 2025) on a curated short audio ($\leq$ 30 seconds) subset of our benchmark named BLAB-MINI.

### 3.2 EVALUATION STRATEGY

**Setup**    Localization, counting, and entire duration tasks include 200 audio samples each, paired with 20 handcrafted, paraphrased questions. Each question is randomly paired with  10 audio samples to ensure diversity. For event duration and emotion tasks, each question is unique to an audio sample as they contain the event information present in the audio, resulting in 200 unique questions.

In our experiments, the audio LMs take a text prompt (instruction) and an audio file as input and produce text as output. In order to ensure that models truly understand the audio samples and avoid biases by providing options, we restrict our benchmark to free-form generation, except for emotion tasks. Also, prior work suggested that multiple-choice question answering is not always reliable, as distractor options are often either too plausible or models exploit shortcuts to arrive at the correct answer (Balepur et al., 2025). For emotion, confounders are generated by extracting plausible, contextually relevant answers from the audio.

| Task | Metric ($\uparrow$) | Gemini 2.0 Flash | Gemini 2.0 Pro |
|---|---|---|---|
| Word Localization | word $F_1$ | 1.12 | 0.19 |
| Advertisement localization | Frame-level $F_1$ | 4.93 | 0.15 |
| NE Localization | Frame-level $F_1$ | 2.97 | 2.14 |
| Speaker Number Estimation | EMA | 8.00 | 8.50 |
| Valence and Arousal Ranking | EMA | 26.28 | 32.00 |
| Emotion Reasoning | EMA | 54.54 | 64.29 |
| Entire Duration | EMA (without / with $\pm 2$ seconds offset) | 0.50/3.50 | 0.00/2.50 |
| Event Duration | EMA (with / without $\pm 2$ seconds offset) | 1.49/4.95 | 1.49/3.96 |

Table 1: Performance comparison of Gemini audio LMs across all BLAB long audio tasks. Both models exhibit similar performance, generally achieving low performance across tasks.

**Prompt Formatting**    To ensure consistent outputs across different inputs and models, we append task-specific suffixes to the original questions. For localization tasks, we instruct the model to return JSON-formatted strings with start and end timestamps. For duration and counting tasks, models are instructed to output a number only, without additional explanation. In emotion tasks, which follow a multiple-choice format, the model is prompted to select the most appropriate option from the provided choices. We provide more details about our prompt formatting in Table 5 in the Appendix.

**Metrics**    We use task-specific metrics to evaluate model performance on BLAB. Model outputs are post-processed to match the expected ground truth format. For localization tasks, we compute Frame-level $F_1$ (*Named Entity & Advertisement*), and Word-level (*Word* ) $F_1$ scores. Counting, duration and emotion tasks are evaluated using exact match accuracy (EMA). For duration tasks, we also report EMA scores with an offset of $\pm 2$ seconds to account for minor timing discrepancies. More details about our evaluation metrics are given in Appendix B.2.

## 4    RESULTS AND DISCUSSION

In Table 1, we present the performance of Gemini 2.0 Flash and Gemini 2.0 Pro on BLAB.

### 4.1    LOCALIZATION TASKS

Word localization appears the most challenging task in BLAB with both models performing extremely poorly. Both Gemini models achieved $F_1$ scores below 2%. These scores are particularly noteworthy, as state-of-the-art word timing models typically achieve scores close to 99% on these last two metrics, as noted by Sainath et al. (2020). We note that each audio sample for this task contains an average of ~10200 words. Gemini, due to its limited output context length of 8096 tokens, is able to generate only ~261 word timestamps per sample, accounting for only about 2% of the ground truth. Gemini 2.0 flash achieves a precision of 24.37%, indicating that the model predictions are correct approximately 24.58% of the time. However, precision for Gemini pro is very low at 3.42%.

Models also perform poorly on NE and advertisement localization, with frame-$F_1$ scores below 5%. For NE localization, they detect 27% of ground truth entities, but fail to correctly locate them accurately. For advertisement localization, performance is better when ads are at the beginning of the audio files, and the Gemini models are more accurate at predicting start times than end times. This leads us to hypothesize that the models estimate rather than detect segments with advertisements.

### 4.2    COUNTING TASK

The EMA on speaker number estimation for both models is below 9%. They typically underestimate the number of unique speakers (see Figure 2a) and struggle with overlapping voice. In some cases, we observe overestimation, likely due to the models considering the same speaker at different positions in the audio as distinct. These errors suggest that audio LMs lack the ability to track speakers

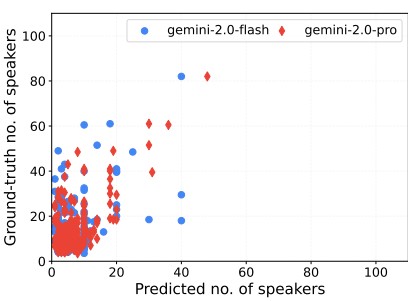
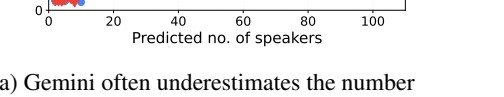
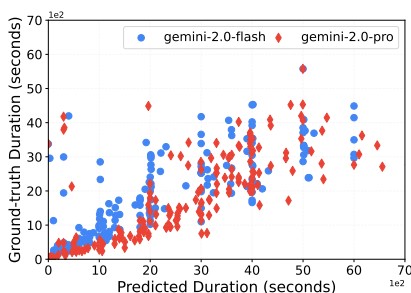

(a) Gemini often underestimates the number of speakers on speaker number estimation

(b) Gemini frequently overestimates the duration of audio samples on entire duration.

Figure 2: Predicted versus groundtruth for speaker number estimation and entire duration

consistently across turns, in conversations with overlapping speech, audio with music, commentary, other forms of extraneous content, or audio with varying prosodic features generally.

### 4.3 DURATION TASKS

**Entire Duration**    We find that Gemini struggles to predict the entire duration on full audio samples (EMA up to 3.50%). Compared to our observations in the speaker number estimation task, our analysis indicates that the models often overestimate duration, as shown in Figure 2b. However, in most cases where the predictions are correct, the actual duration is less than 60 seconds.

**Event Duration**    The performance scores are low as well for this task (EMA up to 4.95%). From our observations, there are no clear trends regarding which acoustic events are predicted more accurately than others. Performance generally varies across different event types, and we observe that the model tends to underestimate event durations more frequently than it overestimates them.

### 4.4 EMOTION TASKS

The highest scores in BLAB are on emotion, with Gemini 2 Pro outperforming Gemini 2 Flash. It achieves 32.00% EMA on emotion ranking and 63.63% on emotion reasoning. For emotion ranking, we find that even Gemini 2 Pro struggles to correctly rank scenarios with extremely calm emotions, often misclassifying them as neutral or highly aroused. However, it accurately predicts higher arousal in 80% of cases. Valence rankings show no clear trends, with accuracy near chance.

## 5 ANALYSIS

**What role does the duration of audio play?**    In this section, we analyze the impact of audio duration on task performance by conducting experiments on shorter audio. This lets us evaluate a broader set of LMs, in particular open-weights models that process audio inputs with a maximum of 30 seconds (Qwen 2, Chu et al., 2024, and Phi-4-Multimodal Instruct, Microsoft et al., 2025) and 5 minutes (Audio Flamingo, Ghosh et al., 2025, and GPT-4o, OpenAI et al., 2024). We conduct analyses on word and entity localization, speaker number estimation and duration tasks. We derive the data from samples in BLAB by extracting audio segments up to 30 seconds. We reuse pre-existing annotations for the localization task and re-annotate the segments for speaker number estimation and

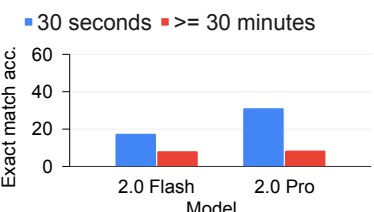

Figure 3: Comparison of long audio and short audio results on Gemini.

duration tasks using the same procedure described in §2. We refer to this as BLAB-MINI. It contains 813 questions and 346 minutes of audio in total. More details are provided in Appendix Table 6.

| Task | Metric ↑ | G2 Flash | G2 Pro | Q2 | AF2 | Phi-4 | GPT-4o |
|------|----------|----------|--------|-----|------|-------|--------|
| Word Localization | Word $F_1$ | **30.22** | 8.61 | 2.43 | – | 2.73 | – |
| NE localization | Frame-level $F_1$ | 45.49 | **49.58** | 12.07 | – | 7.63 | – |
| Speaker Number Estimation | EMA | 17.50 | **31.00** | 7.0 | 6.00 | 15.50 | 14.50 |
| Entire Duration | EMA | 5.00/31.00 | 3.50/**34.50** | 6.5/27.5 | 2.5/20.50 | 3.50/22.00 | **7.00**/27.00 |
| Event Duration | EMA | 9.45/**36.22** | 4.72/29.13 | 3.15/18.90 | 1.57/16.54 | 3.9/24.21 | 1.57/18.11 |

Table 2: Performance comparison of audio LMs on BLAB-MINI audio tasks (≤ 30 seconds). **G2** = Gemini 2.0, **Q2** = Qwen 2.0, **AF2** = Audio Flamingo 2. Gemini outperforms all others. Audio Flamingo 2 and GPT-4o refuse to perform any localization task, so we leave them blank. For event duration, we report scores without and with ±2 seconds offset.

The results are summarized in Table 2 in the main paper and Figure 6 in the appendix. We display comparisons for speaker number estimation in (Figure 3). Gemini models consistently improve across all tasks as the duration is limited to 30 seconds, with the most visible gains observed in word and NE localization. Meanwhile, Table 2 presents a comparison of model performance on BLAB-MINI across multiple models, demonstrating that Gemini outperforms all others on every task.

**Are audio LMs long-form zero-shot reasoners?**     So far, we query the model to directly generate the answer. Inspired by test-time compute research (Kojima et al., 2022; Wei et al., 2022), we explore zero-shot chain-of-thought approaches. We append an auxiliary *reasoning prompt* to the original prompt to guide the model in generating reasoning chains that could lead to better predictions. We test the following prompts that have been effective in text-only LMs: *"Let's think step by step"* and *"Explain your reasoning before making a prediction"*. We conduct this analysis on speaker number estimation and event duration with Gemini 2.0 Flash and use all long audio examples in BLAB.

For event duration, we observed accuracies of 6.93 and 6.44 with the reasoning prompts *Let's think step by step"* and *Explain your reasoning before making a prediction"*, respectively—an average improvement of 3% over the original accuracy of 3.96% without any reasoning prompts (cf. Table 1). For speaker number estimation, the accuracies with both reasoning prompts are 9.55% and 11% respectively, a slight performance boost compared to the original 8% accuracy.

We analyzed individual reasoning steps generated for both prompts, and observed that the model excels at planning but struggles with correctly executing each sub-task in its plan due to limited auditory understanding. For instance, in Table 4 in the Appendix, we can see that both reasoning prompts contain task decomposition, but sub-task results are often incorrect due to weak audio perception. While reasoning traces are coherent, limited auditory grounding leads to errors in the final output. Audio LMs are therefore not truly zero-shot reasoners, and more advanced test-time strategies may be needed to better align reasoning with accurate auditory perception.

**Robustness of audio LMs on long-form audio**    Recent work has suggested that certain audio LMs rely only on text prompts, rather than jointly attending to text and audio input. Sakshi et al. (2024) report that GAMA (Ghosh et al., 2024a), Qwen2-Instruct (Chu et al., 2024) and Gemini Pro (Gemini Team et al., 2024) are more robust to noisy audio and are usually more attentive to audio content compared to other models like SALMONN (Tang et al., 2024) . However, these experiments were done on short audio samples (up to 30 seconds).

We follow Sakshi et al. (2024) and start by comparing the original model's prediction for speaker number estimation with its prediction when the audio input is replaced with random Gaussian noise. In a different setting, we also replaced the original audio input with silence. All experiments are performed on Gemini, Phi-4-Mini and Audio

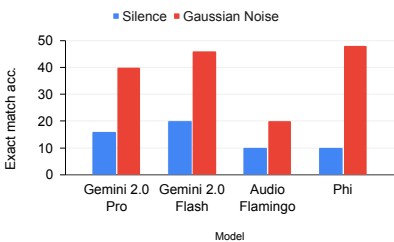

Figure 4: Performance comparison when the original audio input is replaced with silence or Gaussian noise. As the entire input is noisy, the ground truth label is 0

Flamingo, since they support longer durations than other models. We generate 5 minute noisy audio samples for Audio Flamingo and Phi-4-Mini, while we generate one hour long noisy audios samples for Gemini. In contrast to previous work on short audio (Sakshi et al., 2024), which finds that audio (LMs) are robust to noisy short audio samples, our findings reveal a different trend for long noisy inputs. Specifically, our analysis shows in Figure 4 that the models are not robust to noisy inputs, and they are particularly less robust to silence than to Gaussian noise.

Next, we investigated how the positioning of noise affects the robustness of an audio LM. Unlike our previous analysis, where the entire input was replaced with Gaussian noise or silence, we now introduce 30-second audio clips from BLAB-MINI into 60-minute noisy recordings for Gemini and 5-minute noisy recordings for Phi and Audio Flamingo, placing the clips at various positions. Our goal is to measure the model's ability to disregard background noise and focus on meaningful content. We conducted this experiment for speaker number estimation, varying the placement of the clean audio clip based on the model's maximum input duration.

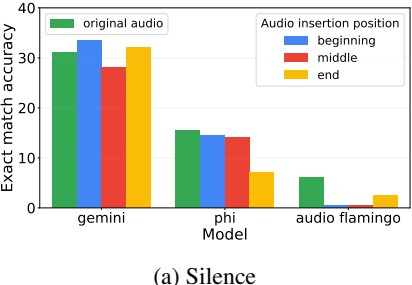

(a) Silence

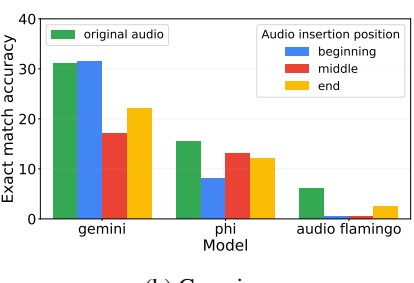

(b) Gaussian

Figure 5: Placing a 30-second clean audio clip at different points within a long, noisy audio input impacts speaker number estimation performance. Proprietary models like Gemini perform better when the clean clip is positioned at the beginning or end of the noisy audio.

In Figure 5, we compare the performance of Gemini, Phi, and Audio Flamingo. Gemini's performance degrades when the 30-second audio clip is placed in the middle, suggesting that the model struggles to effectively use middle information in long input contexts. This is consistent with previous findings on text LMs, where performance peaks if relevant information is at the beginning or end and significantly drops when it is in the middle, even for models designed to handle long contexts (Liu et al., 2024). Degradation is worse with Gaussian noise than silence. For 5-minute inputs, Phi performs better when the 30-second clip is placed in the middle of noise, while Audio Flamingo shows consistent degradation, likely due to difficulty distinguishing noise from actual signal as seen in Figure 4.

## 6 CONCLUSION

In this paper, we introduce Brutally Long Audio Bench (BLAB), a challenging benchmark for evaluating long-form audio understanding and reasoning in audio language models across localization, duration estimation, emotion, and counting tasks. BLAB is the first benchmark to assess audio LMs on long-form audio, with durations ranging from over 15 minutes to up to 2 hours, with tasks that are both practical and readily applicable to real-world use cases. Our evaluation of six open-weight and proprietary audio LMs reveals that these models struggle substantially with long-form speech, with performance deteriorating as audio duration increases. Additionally, we find that audio LMs perform poorly on both temporal and counting tasks and struggle to process non-phonemic information in audio. Contrary to existing findings on short-form audio, our analysis suggests that audio LMs are not particularly robust when handling long-form speech. These models tend to rely more on prompts than on actual audio content, making them susceptible to distractions from noisy audio, such as Gaussian noise and silence. We provide a detailed report of our data curation and evaluation framework. Overall our findings underscore the need for more approaches to developing long-context multimodal language models with strong long-form audio understanding capabilities.

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

# A APPENDIX

## LIMITATIONS

Audio data for BLAB is sourced from various real-world recordings, including interviews, podcasts, and political speeches. Our annotation framework leverages human-AI collaboration. However, we observe that overlapping speech is common, which can impact the accuracy of automatic annotation tools like WhisperX and Pyannote (Bain et al., 2023; Bredin, 2023), as they may not be robust in handling such cases. To address this, our framework includes human verification steps to improve annotation quality.

## ETHICS STATEMENT

BLAB is entirely sourced from Creative Commons-licensed videos on YouTube, using a rigorous human-assisted filtering process to ensure diverse and high-quality content. Human speech is a particularly sensitive modality, as it is deeply personal and can convey not only language but also emotions and mental states. Each audio file in BLAB is carefully selected, with deeply sensitive material excluded to protect privacy. We have also ensured that no child sexual abuse material is present in our dataset.[1] We recognize that speech recordings can be used to track or identify individuals without their knowledge or consent. To address this, we have built our dataset using publicly available YouTube data that complies with ethical guidelines regarding privacy and data usage. However, we also acknowledge the potential risks of data misuse, such as the unintended identification of individuals or the reinforcement of biases in downstream audio language models due to potential contamination. Therefore, we encourage responsible use of our dataset and highlight the importance of considering privacy and ethical concerns when applying it to evaluate real-world applications. To promote transparency and reproducibility, we will make our benchmark publicly available, along with associated evaluation metrics and data curation framework, allowing the research community to contribute and build upon our work.

# B RELATED WORK

**Audio Benchmarking**   Audio benchmarks can be broadly classified into two main categories based on their scope and purpose: *Task-Specific Benchmarks* focus on evaluating models' performance on particular audio tasks. Examples include text-audio retrieval (Koepke et al., 2021), compositional audio reasoning (Ghosh et al., 2023), automatic speech recognition (Panayotov et al., 2015; Shi et al., 2024), audio captioning (Drossos et al., 2019; Kim et al., 2019), and emotion recognition (Livingstone & Russo, 2018). Several benchmarks combine such tasks into a collection such as SUPERB (Yang et al., 2021), HEAR (Turian et al., 2022), among others. Our work falls under the umbrella of *instruction following benchmarks* which assess model capabilities to understand audio signals and follow instructions in a conversation format. Dynamic-SUPERB was one of the first benchmarks of this kind (Huang et al., 2023), followed by AIR-Bench (Yang et al., 2024) and MMAU (Sakshi et al., 2024). However, almost all of these benchmarks contain samples with a maximum duration of 30 seconds. Most closely related to our work is Ghosh et al. (2025), who release LongAudioBench containing tasks with samples with a maximum duration of 5 minutes. There are existing long-form speech benchmarks such as CHiME Barker et al. (2018); Watanabe et al. (2020), AMI Carletta et al. (2005), and TEDHernandez et al. (2018), which often span more than one hour and primarily focus on speech recognition. Our work focuses on much longer long audio samples which can be up to 2 hours long with a large focus on audio reasoning.

**Long Context Benchmarks**   As the context length for text-based LMs has increased, new benchmarks have emerged to test how well the models handle long-form text. These include tasks like information retrieval (Needle-in-a-Haystack; Nelson et al., 2024) and synthetic long-range reasoning (Long Range Arena; Tay et al., 2020). Newer benchmarks such as LongBench (Bai et al., 2023) evaluate models on tasks over inputs with up to 128K tokens. While these efforts have advanced

---

[1]https://www.missingkids.org/cybertiplinedata

long-context evaluation in text, similar benchmarks are largely missing for audio. Our work fills this gap by introducing a benchmark for evaluating models on extremely long-form audio understanding.

| Task | TD (hrs.) | AD (mins.) | Question Example |
|---|---|---|---|
| Word Localization | 191 | 57 | Align the provided audio file with each word in its transcript. For each word spoken, predict the start and end timestamps in seconds and milliseconds. |
| NE Localization | 110 | 56 | Your task is to analyze an audio file and detect all **Movie** named entities present within it. **Movie** entity refers to the name of **feature films or animated movies**. For each detected **Movie** entity, provide the **start** and **end** timestamps (in seconds and milliseconds) that indicate the location and duration of the entity in the audio. |
| Advertisement Localization | 232 | 70 | Given the audio file, your task is to detect the presence of promotions and advertisements within the audio. You are required to predict the start and end timestamps (in seconds and milliseconds) for the entire span for all promotion and advertisement segments you have identified. |
| Speaker Number Estimation | 176 | 53 | What's the number of unique voices detected in this sound file? |
| Valence & Arousal Ranking | 18 | 39 | How would you assess the valence of the speaker in the fourth apology clip within this audio file? |
| Emotion Reasoning | 16 | 44 | Throughout the course of this debate, which speaker demonstrates the most consistent emotional intensity, particularly in their tone and delivery? |
| Entire Duration | 92 | 27 | What is the total duration of this audio file in seconds only? |
| Event Duration | 174 | 51 | A harmonica is being played fifteen seconds towards the end of this audio. How long in seconds was this instrument played for ? |

Table 3: Statistics and examples of questions for each task in BLAB. TD and AD represent total and average duration, respectively. Each task consists of 200 question-audio-answers.

| Question | Ground Truth | Let's think step by step | Explain your reasoning |
|---|---|---|---|
| What is the total time in seconds that the first panelist spoke for throughout the entire podcast? | 1160 | Here's how we can calculate the total speaking time for the first panelist:1. **Identify the first panelist:** The first panelist is Dan Mogulof.2. **Locate the start and end times of Dan's first speaking segment:** Dan starts speaking at [00:02:00] and ends at [00:14:50].3. **Calculate the duration of Dan's first speaking segment:** 14 minutes and 50 seconds − 2 minutes and 0 seconds = 12 minutes and 50 seconds.4. **Locate the start and end times of Dan's second speaking segment:** Dan starts speaking at [00:48:30] and ends at [00:56:30].5. **Calculate the duration of Dan's second speaking segment:** 56 minutes and 30 seconds − 48 minutes and 30 seconds = 8 minutes and 0 seconds.6. **Convert the durations to seconds:** * 12 minutes and 50 seconds = (12 * 60) + 50 = 720 + 50 = 770 seconds * 8 minutes and 0 seconds = (8 * 60) + 0 = 480 seconds7. **Add the durations together:** 770 seconds + 480 seconds = 1250 seconds.Therefore, the first panelist, Dan Mogulof, spoke for a total of 1250 seconds throughout the entire podcast. | Here's how we can estimate the speaking time of the first panelist:1. **Identify the Speaker:** The first panelist is Dan Mogulof.2. **Locate the Start Time:** The first panelist begins speaking at 00:01:55.3. **Locate the End Time:** The first panelist stops speaking at 00:18:00.4. **Calculate the Duration:** 00:18:00 − 00:01:55 = 00:16:05.5. **Convert to Seconds:** (16 minutes * 60 seconds/minute) + 5 seconds = 965 seconds.**Prediction:** The first panelist spoke for approximately 965 seconds. |

Table 4: Comparison of reasoning prompts and model responses for estimating the duration of a person speaking in a panel discussion. While the model effectively decomposes the task and generates relevant reasoning, it often lacks the auditory understanding needed to execute sub-tasks accurately.

## B.1    PROMPT FORMATTING

| Task | Prompt Format |
|---|---|
| Word Localization | Align the provided audio file with each word in its transcript.  For each word spoken, predict the start and end timestamps in seconds and milliseconds.

Format the response as:
{{'word':  '<word>', 'start':  <start timestamp of the word>, 'end':  <end timestamp of the word>}}} |
| NE Localization | Your task is to analyze an audio file and detect all **movie** named entities present within it.  **Movie** entity refers to the name of **feature films or animated movies**. For each detected **movie** entity, provide the **start** and **end** timestamps (in seconds and milliseconds) that indicate the location and duration of the entity in the audio.

**Expected Output Format**
Your response should be formatted as a list of dictionaries:
{{ [ 'entity':  '<movie name>', 'category':  'movie', 'start':  <start timestamp>, 'end':  <end timestamp>, 'entity':  '<movie name>', 'category':  'movie', 'start':  <start timestamp>, 'end':  <end timestamp> ] }}} |
| Advertisement Localization | Given the audio file, your task is to detect the presence of promotions and advertisements within the audio.  You are required to predict the start and end timestamps (in seconds and milliseconds) for the entire span for all promotion and advertisement segments you have identified.

 Format the response as:
{{ "advertisement":  <advertisement text>, "start":  <start timestamp of the advertisement>, "end":  <end timestamp of the advertisement>}}} |
| Speaker Number Estimation | What's the number of unique voices detected in this sound file?

Provide only the numeric value without any explanation. |
| Continued on next page | |

**Table 5 – continued from previous page**

| Task | Prompt Format |
|------|---------------|
| Valence & Arousal Ranking | How would you assess the valence of the speaker in the fourth apology clip within this audio file?

Listen to the audio and select one option from the provided choices that best matches the answer.  Return only that option.
Options:
(A) Very Pleasant
(B) Pleasant
(C) Neutral
(D) Very Unpleasant
(D) Unpleasant |
| Emotion Reasoning | Throughout the course of this debate, which speaker demonstrates the most consistent emotional intensity, particularly in their tone and delivery?

Listen to the audio and select one option from the provided choices that best matches the answer.  Return only that option. Options:
(A) The Tory Party leader
(B) The Labour Party leader
(C) Both speakers exhibit similar levels
(D) It is difficult to determine |
| Entire Duration | What is the total duration of this audio file in seconds only?

Provide only the numeric value without any explanation. |
| Event Duration | A harmonica is being played fifteen seconds towards the end of this audio. How long in seconds was this instrument played for ? Provide only the numeric value as an integer without any explanation.  Do not use the MM:SS format. |

Table 5: Exact prompt formats used for evaluating each task in BLAB.

## B.2 METRICS

Given a model output, we post-process it to match the expected ground-truth format. For localization tasks, we use the `json_repair` library.[2] For the remaining tasks with numeric outputs, we use regular expressions to extract relevant numerical values. The regular expression are designed to identify and format various numeric formats, including integers and numbers expressed with units (e.g., "35 seconds" or 00:23:00).

**Localization Tasks**  For these tasks, the models are expected to generate JSON outputs. Word localization is typically evaluated using metrics that compare start and end timestamp differences for matching words in the audio transcript, detecting delays in word onset or offset Sainath et al. (2020). For ease of evaluation, we report $F_1$ scores on the number of correctly aligned words.

For NE and advertisement localization, which are span-localization tasks, we use Frame-level $F_1$ as mentioned in Shon et al. (2023). This metric is derived from question-answering evaluation frameworks (Chuang et al., 2020), and measures the overlap between the predicted and the ground truth answer spans.

**Counting and Duration and Emotion tasks**  These tasks require numeric answers and are evaluated using exact match accuracy (EMA).

## B.3 DATASET STATISTICS OF BLAB-MINI

BLAB-MINI is a subset of BLABthat contains audio samples less than or equal to 30 seconds of audio on average.

| Task | TD (mins.) | AD (seconds.) | Number of questions |
|---|---|---|---|
| Word Localization | 89.00 | 30.00 | 178 |
| NE localization | 36.00 | 30.00 | 107 |
| Speaker Number Estimation | 99.17 | 30.00 | 200 |
| Entire Duration | 58.78 | 17.64 | 200 |
| Event Duration | 63.55 | 29.79 | 128 |
| **Total** | **346.5** | | **813** |

Table 6: Statistics and examples of questions for each task in BLAB-MINI. TD and AD represent total and average duration, respectively.

---

[2]https://pypi.org/project/json-repair/

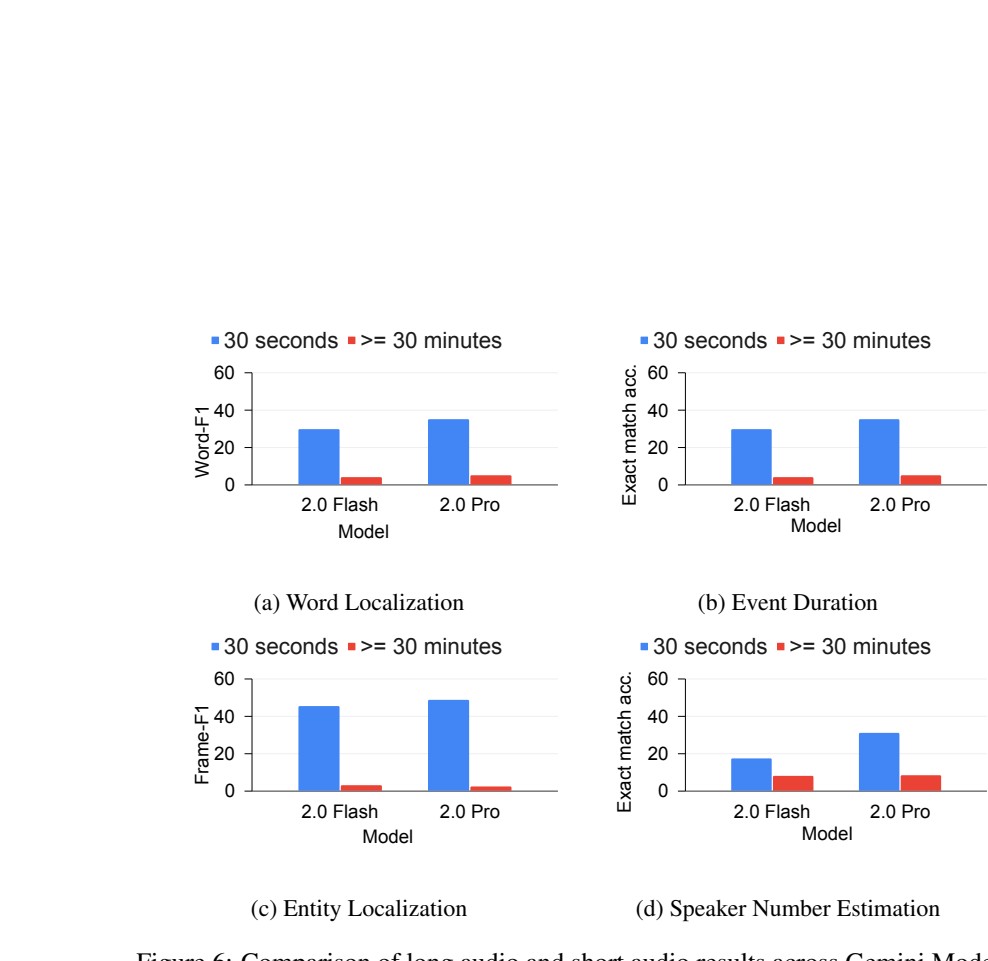

(a) Word Localization

(b) Event Duration

(c) Entity Localization

(d) Speaker Number Estimation

Figure 6: Comparison of long audio and short audio results across Gemini Models

