# OpenReview forum: "BLAB: Brutally Long Audio Bench"
_ICLR.cc/2026/Conference — Submitted to ICLR 2026_

### Official Review · Reviewer_1ctL · 2025-11-01

**Soundness:** 3
**Presentation:** 3
**Contribution:** 2
**Rating:** 4
**Confidence:** 4

**Summary:**

A  benchmark for large audio language models, with speech-focused tasks including temporal localization  of words and named entities,  emotion, speaker counting and event duration. Due to the incapability of existing models to handle large audio, the evaluation is on two Gemini models, with a shorter version of the dataset used for broader evaluation. Several newer models have not been benchmarked (Qwen 2.5 Omni, Audio Flamingo 2, Audio Flamingo 3, Kimi.)  Moreover, the benchmark does not appear to include music. Nevertheless, it could be a valuable contribution to benchmarking audio models, if details of its release, and the ease of benchmarking (scripts, frameworks) were discussed in detail.

**Strengths:**

Many practical tasks for LALMs require diarization, event or named entity recognition, and these are tasks at which current models do poorly. A public benchmark focused on these would be valuable.

**Weaknesses:**

The limited number of models studied.
The lack of a commitment to release the benchmark and associated scripts.
Does not cover music or entertainment audio -- is speech/noise focused.

**Questions:**

which audio flamingo model did you evaluate?
A comparison of the models' performance on Chime with their performance on BLAB would be interesting to see what value BLAB adds.
What languages were covered?

---

> ### Author Response · Authors · 2025-11-21
>
> We thank the reviewer for their feedback and time, and for highlighting the value of our work on long-form evaluation of audio LMs. We provide clarifications and responses to all your comments.
>
> **Limited number of models studied**
> - BLAB’s primary goal is to evaluate LALMs on long-form audio. We mentioned in the paper on lines [248 - 254] that most existing audio LMs can only process short audio inputs: GPT-4o and AudioFlamingo handle ~10 minutes, and others like Qwen typically handle ~30 seconds. BLAB’s average audio length is 51 minutes, which makes evaluating these models infeasible.
>
> - For this reason, all long-audio evaluations use Gemini, the only model that supports more than 50 minutes of audio, even up to 9 hours. To broaden our analysis, we introduced BLAB-mini, a short audio track (30-second clips) with the same tasks as BLAB. This lets us test a wider range of recent models, including Qwen, AudioFlamingo, Phi, and GPT4o.
>
> - We have conducted additional experiments on newer models across the short-audio track, and we observe some performance improvements. Qwen 3 shows notable gains, though some of this is likely due to the increase in model size (Qwen2 7B → Qwen3 30B). Overall, our earlier conclusion still remains valid: Gemini remains the strongest model across tasks, while still leaving substantial room for improvements.
> - On the long audio track, we also see improvements across most tasks going from Gemini 2.0 to Gemini 2.5 families; however, we note that there is still also much headroom for improvements, especially on the lowest-performing tasks.
>
> **Short Audio (Higher is better for all tasks)**
>
> | Model               | Word Localization (F1) | NE Localization (F1) | Speaker Num. Estimation (Acc) | Event duration (Acc) | Entire Duration (Acc) |
> |--------------------|-----------------------|---------------------|-------------------------------|--------------------|----------------------|
> | Audio Flamingo 2    | 0                     | 0                   | 6.00                          | 1.57               | 2.5                  |
> | Audio Flamingo 3    | 0                     | 0                   | 2.00                          | 5.47               | 2.50                 |
> | Qwen2audio          | 2.43                  | 12.07               | 7.00                          | 3.15               | 6.5                  |
> | Qwen3omni           | 4.26                  | 16.24               | 16.41                         | 13.28              | 29.50                |
> | Phi-4               | 2.73                  | 7.63                | 15.5                          | 3.9                | 3.50                 |
> | GPT-4o              | 0                     | 0                   | 14.5                          | 1.57               | 7.00                 |
> | Gemini 2.0 Pro      | 8.61                  | 49.58               | 31                            | 4.72               | 3.50                 |
> | Gemini 2.0 Flash    | 30.22                 | 45.49               | 17.5                          | 9.45               | 5.00                 |
> | Gemini 2.5 Pro      | **66.37**             | **69.94**           | **35.35**                     | **10.16**          | 1.50                 |
> | Gemini 2.5 Flash    | 29.16                 | 52.36               | 26.77                         | 2.38               | **4.0**              |
>
> **Long Audio (Higher is better for all tasks)**
>
> | Model               | Word Localization (F1) | NE Localization (F1) | Advertisement Localization | Speaker Num. Estimation (Acc) | Event duration (Acc) | Entire Duration (Acc) | Emotion ranking (Acc) | Emotion reasoning (Acc) |
> |--------------------|-----------------------|---------------------|----------------------------|-------------------------------|--------------------|----------------------|----------------------|------------------------|
> | Gemini 2.0 Flash    | 1.12                  | 2.97                | 4.93                       | 8 .00                            | 1.49               | 0.50                 | 26.28                | 54.54                  |
> | Gemini 2.0 Pro      | 0.19                  | 2.14                | 0.15                       | 8.50                           | 1.49               | 0.00                 | 32.00                | 64.29                  |
> | Gemini 2.5 Pro      | **15.79**             | **11.00**              | **11.48**                  | **35.18**                     | 1.98               | 1.50                  | 25.00                | **66.00**                 |
> | Gemini 2.5 Flash    | 6.16                  | 1.93                | 17.8                       | 14.07                         | **2.48**           | 0.50                 | 19.2                 | 56.8                   |

---

> > ### Author Response · Authors · 2025-11-21
> >
> > **The lack of commitment to release the benchmark and associated scripts**
> > - We will release the full benchmark, along with the code and model generations, to support reproducibility and further research. Open access to BLAB is central to enabling the community to build on our findings.
> >
> > **Benchmark does not cover music and entertainment audio**
> > - We agree that a broader range of audio domains would be valuable. Our focus was on realistic, understandable tasks we could reliably evaluate using long-form inputs. It was not feasible to directly include long-form music or full-length entertainment media due to licensing and data availability constraints. However, we note that these domains are still indirectly represented in our data sources. For example, several samples come from movie-review videos that include conversational speech, background sounds, and brief music segments. While BLAB does not explicitly cover isolated music or entertainment content, it does capture mixed audio conditions that reflect aspects of these domains as suggested.
> >
> > **Which AudioFlamingo model did you evaluate?**
> > - In the submitted draft, we reported results for AudioFlamingo 2. Since then, we have run additional experiments and now also include results for AudioFlamingo 3.
> >
> >
> > **Comparison of models performance on Chime**
> > - CHiME is an annual shared task with multiple tasks that are entirely different from the tasks in BLAB. In our submitted draft, we cited CHiME as a complementary source of long-form audio, but we do not believe that evaluating models on CHiME would provide results comparable to the tasks we focus on. Does the reviewer have suggestions for specific tasks in CHiME that might be relevant to compare to?
> >
> > **What languages were covered**
> > - BLAB currently includes only English data. We made this choice because most existing audio LMs are trained primarily in English. We hope our work highlights this limitation and motivates new research directions toward multilingual long-audio understanding.  We will clarify this in the next version of the paper.

---

### Official Review · Reviewer_NZWq · 2025-11-04

**Soundness:** 3
**Presentation:** 3
**Contribution:** 3
**Rating:** 6
**Confidence:** 3

**Summary:**

This paper introduces BLAB, a new benchmark designed to evaluate the reasoning capabilities of audio language models on long-form audio. The authors argue that existing benchmarks, which focus on short clips, fail to assess a critical aspect of real-world audio understanding. BLAB consists of over 833 hours of diverse audio, with an average clip length of 51 minutes. It features 8 challenging tasks across 4 categories: localization, counting, emotion, and duration. The authors evaluate several audio LMs and find that all models "struggle" substantially with these long-context tasks.

**Strengths:**

- A Novel Benchmark: The paper's primary contribution is the BLAB benchmark itself. It addresses a critical, well-documented gap in the field: the lack of evaluation for audio-grounded reasoning on long-form content (averaging 51 minutes). This moves the community beyond short-clip evaluations to a more realistic and challenging domain.
- Transparent Data Collection Pipeline: The authors detail a rigorous data pipeline, using permissively licensed sources from YouTube. This process is strengthened by a "human-assisted filtering procedure" and a human-AI collaborative annotation framework to ensure data quality.
- Insightful Analysis: The paper provides valuable insights by evaluating state-of-the-art models and identifying key failure modes. These include clear performance degradation as audio duration increases and the presence of the "lost in the middle" problem for audio inputs.

**Weaknesses:**

- The evaluation of the full, long-form benchmark (BLAB) relies almost entirely on closed-source, proprietary models like Gemini. While this effectively demonstrates the benchmark's difficulty, it limits reproducibility and prevents the research community from conducting a deeper analysis of the models' failure modes. The open-source models were only evaluated on the BLAB-MINI subset, so their long-context audio capabilities remain unevaluated.
- The design of the word localization task seems confounded by model limitations that are unrelated to audio reasoning. The paper notes that the model's poor performance is largely due to its output token limit (8096), which only allows it to generate ~2% of the ground truth timestamps for an average audio file. This makes it an unfair evaluation, as it is testing the model's output length rather than its true localization ability.

**Questions:**

Please refer to the weaknesses above for the questions.

---

> ### Author Response · Authors · 2025-11-21
>
> We thank the reviewer for their feedback and for recognizing the strengths of our work, noting BLAB as a novel benchmark for long-form audio, our transparent data collection pipeline, and the insights into model limitations such as performance degradation over long inputs.
>
> **Evaluation of long-form audio is only done on Gemini**
> - This is a valid point. We mentioned in the paper in lines [248 - 254] that most existing audio LMs can only process short audio inputs: GPT-4o and AudioFlamingo handle ~10 minutes, and others like Qwen typically handle ~30 seconds.  In fact, the longest any open-source model can currently process is about 10 minutes. BLAB’s average audio length is 51 minutes, which makes evaluating these models infeasible. This is why all long-audio evaluations use Gemini, the only model that supports more than 50 minutes of audio, even up to 9 hours.
> The challenge of reproducing results on closed frontier models also highlights the need for open-source LALMs capable of handling long-form audio.
>
>
> **Word localization task is confounded with model limitations**
> - Thanks for pointing this out. We agree with you that model performance is limited by the maximum output token length. However, we believe that this highlights gaps in end-to-end audio LMs for performing long-form audio tasks requiring long output generation.  For a fairer comparison, we also computed the precision of predicted word timestamps and still observed low scores (6.162 for Gemini Flash and 15.793 for Gemini Pro), indicating that even when the model predicts timestamps up to its context limit, its accuracy is still very low. We will add these explanations to the final draft.

---

### Official Review · Reviewer_pRew · 2025-11-04

**Soundness:** 3
**Presentation:** 2
**Contribution:** 3
**Rating:** 6
**Confidence:** 4

**Summary:**

**Summary:** Audio language models aim to understand all the spoken interactions within a single recording. However, most audio language models can only process short audio segments (under 30 seconds), and they are rarely evaluated on longer, conversational speech segments.

Authors of this work developed a benchmark that consists of very long audio segments, and used it to evaluate some SOTA models' performance, and showed their limitations at this scale and how they struggle differently on diverse tasks due to such much longer audio recordings.

To be specific, the benchmark (BLAB) is 833 hours of conversational speech, with 8 tasks, and evaluates 4 reasoning skills, which are temporal localization, speaker counting, emotion interpretation, and duration estimation. Each audio recording has human-annotated question and answer pairs. A total of 6 audio LMs are evaluated on BLAB.

**Strengths:**

**Strength:**
The background of the limitations of current audio language models and the motivations are clearly and logically stated and emphasized throughout the paper.

Each category of the reasoning capabilities and the tasks under it are clearly designed and described in detail.

The experimental setup is designed carefully and fairly.

**Weaknesses:**

**Weakness:**

The description of the tasks contained in BLAB and the 4 reasoning skills can be clearer. For instance, after reading the abstract section, and up to the point 'across eight tasks and evaluates four fundamental reasoning skills', the relationship between the eight tasks and the four reasoning skills can be a little confusing. It would be better if you could describe that the 8 tasks you are evaluating are under 4 categories, just like your caption for figure 1, in the abstract, to avoid this minor confusion.

It is also unclear whether each audio segment has the 8 tasks, or if they have variations.

At the end of the Introduction section, you briefly mentioned the audio LMs' performance on a certain category, such as localization. It looks like the reported metric. I can only assume that all 3 tasks in this category are classification tasks, so the reported results are the average? Since you discussed the performance here, this needs to be better explained.

After evaluation, it is found that prompts have higher importance than the actual audio contents. If so, I would suggest placing a greater emphasis on various types of prompts, beyond those introduced on page 8, to better understand the limitations of the prompts.

For fairness and consistency, I understand that authors are only focused on audio sources on YouTube. But if the authors have resources and time, I would suggest additional evaluations on more specific tasks, such as speech datasets that are specifically for counting, emotion recognition, etc. There are some datasets that have conversation speech that far exceeds 30 seconds. So the results evaluated on the current audio LMs could be different, then you can at least have some conclusions based on the difference between BLAB and these datasets. Because focusing only on YouTube and English datasets is also a limitation.

**Questions:**

For emotion recognition, you can take a look at the work below that also aligns with your approach and considerations somehow.

"The MERSA dataset and a transformer-based approach for speech emotion recognition." Proceedings of the 62nd Annual Meeting of the Association for Computational Linguistics (Volume 1: Long Papers). 2024.

---

> ### Author Response · Authors · 2025-11-21
>
> We thank the reviewer for their feedback and for recognizing the strengths of our work, including the clear presentation of LALM limitations and motivations, the well-designed reasoning tasks, and the careful experimental setup.
>
> **The description of the tasks contained in BLAB and the 4 reasoning skills can be clearer.**
> - BLAB consists of eight different tasks in total: (Named Entity Localization, Word Localization, Advertisement Localization, Speaker Number Estimation, Event duration, Entire duration, Valence and arousal Ranking, and Emotion Reasoning ). As described in the paper, each of these tasks falls under one of four broader categories: localization, duration estimation, counting, and emotion interpretation.
> The relationship between these task categories and the four reasoning skills could be clearer in the current draft. For simplicity, we grouped the underlying reasoning skills into temporal localization, duration estimation, speaker counting, and emotion interpretation. However, these skills are complementary rather than isolated. For example, speaker counting inherently relies on temporal localization to track speaker changes across long audio.
> We will revise the relevant sections in the paper draft to make this relationship explicit and clarify how each task aligns with the reasoning skills evaluated in the benchmark.
>
>
> **It is also unclear whether each audio segment has the 8 tasks, or if they have variations.**
> - As noted in Line 108 of the draft, each task in BLAB is evaluated on its own set of audio files. We use 200 unique audio files per task, except for the two emotion tasks, which share 200 files in total. This means audio files are not annotated for all eight tasks simultaneously; instead, each task has its own dedicated set of long-form audio samples.
>
> **After evaluation, it is found that prompts have higher importance than the actual audio contents, it is found that prompts have higher importance than the actual audio contents. If so, I would suggest placing a greater emphasis on various types of prompts.**
> - Our results do show that models often rely more on text priors from the prompts than on the audio itself.
> We agree that including prompt variations is important. In the current draft, we explored this in detail for Speaker Number Estimation, using different prompts and replacing the audio with different forms of noise. In preliminary analysis, we also saw consistent results across tasks. In the final version, we will explicitly clarify these findings.
>
> **Additional evaluations on more specific tasks, such as speech datasets that are specifically for counting, emotion recognition**
> - Thank you for the suggestion. While datasets like MERSA are useful for counting or emotion recognition, they are short-form, averaging only 6 seconds, and not aligned with our focus on long-audio reasoning. Existing benchmarks like MMAU and AirBench already cover many of these short-audio tasks using such sources. For BLAB, we curated and annotated all data from scratch to ensure long duration and consistency with our target tasks.

---

### Official Review · Reviewer_t4yw · 2025-11-06

**Soundness:** 2
**Presentation:** 3
**Contribution:** 2
**Rating:** 2
**Confidence:** 5

**Summary:**

The paper proposes BLAB, a long-form (minutes to hour-scale) "audio LM" benchmark spanning: (i) localization (word, named-entity, advertisement), (ii) counting (speaker number), (iii) duration (file/event), and (iv) emotion (valence/arousal ranking; comparative reasoning). The core datasets are Creative-Commons speech from YouTube; annotation relies heavily on WhisperX forced alignment plus LLM-assisted span extraction; evaluation is mostly exact-match/JSON and frame-F1. Results chiefly report Gemini 2.0 Flash/Pro on long audio; robustness tests replace/overlay content with silence or Gaussian noise; a short-audio subset (BLAB-MINI) is included.

**Strengths:**

- Truly long speech (≈50–60 min avg per item) where current LAMs struggle, including controlled noise/silence robustness and position-sensitivity tests.
- Clear, reproducible prompt formats and metric definitions per task family.

**Weaknesses:**

- Doesn't evaluate many recent models like Gemini 2.5, GPT-4o-audio.
- Doesn't evaluate models recent models like Audio Flamingo 3 which claims long audio understanding.
- NE & Ad localization are derived by running text-only LMs on transcripts, then mapping spans back via WhisperX timestamps. This makes many items solvable from text alone and entangles evaluation quality with ASR/FA errors rather than acoustic understanding.
- Word timestamps over 191 hours are WhisperX with just ~1% corrections, reviewed on a subset by an author is kind of insufficient for long-audio with overlaps; alignment errors propagate directly into and impact ground truth.
- Despite "audio LM" claims, items come from interviews/podcasts/political talks; background music is incidental. There's almost no coverage of non-speech environmental or musical structure. With the recent advancements of large audio language models I think the benchmark should also focus on including other modalities of audio and not just speech.
- How many items per task were fully human-audited? What is inter annotator agreement?

**Questions:**

See weakness section.

---

> ### Author Response · Authors · 2025-11-21
>
> We thank the reviewer for highlighting BLAB’s strengths, including its truly long-form audio, which challenges current LALMs across tasks. Below, we provide clarifications and responses to your comments
>
>  **Evaluation of recent models like Gemini 2.5, GPT4o, and Audio Flamingo 3**:
> - We have included updated results on more recent models in the table below. We would like to point out that Table 2 in our original draft displayed results for GPT4o on short audio. We also noted in lines [250 and 254] that Audio Flamingo 2 at the time processed only up to 5 minutes of audio, and we could only process audio files up to 8 minutes with GPT4o. **Audio Flamingo 3** takes a maximum of 10 minutes of audio. This is why our long-audio evaluations focus on Gemini, the only model that supports more than 50 minutes of audio, since BLAB’s average audio length is 51 minutes.
> - Our earlier conclusions still hold: **Gemini** remains the strongest model across tasks, while still leaving substantial room for improvements. On the long audio track, we also see improvements across most tasks going from Gemini 2.0 to Gemini 2.5 families, however, we note that there is still also much headroom for improvements, especially on the lowest-performing tasks.
>
>
> **Short Audio (Higher is better for all tasks)**
>
> | Model               | Word Localization (F1) | NE Localization (F1) | Speaker Num. Estimation (Acc) | Event duration (Acc) | Entire Duration (Acc) |
> |--------------------|-----------------------|---------------------|-------------------------------|--------------------|----------------------|
> | Audio Flamingo 2    | 0                     | 0                   | 6.00                          | 1.57               | 2.50                 |
> | Audio Flamingo 3    | 0                     | 0                   | 2.00                          | 5.47               | 2.50                 |
> | Qwen2audio          | 2.43                  | 12.07               | 7.00                          | 3.15               | 6.50                  |
> | Qwen3omni           | 4.26                  | 16.24               | 16.41                         | **13.28**             | **29.50**                |
> | Phi-4               | 2.73                  | 7.63                | 15.5 0                        | 3.9                | 3.50                 |
> | GPT-4o              | 0                     | 0                   | 14.5 0                         | 1.57               | 7.00                 |
> | Gemini 2.0 Pro      | 8.61                  | 49.58               | 31.0                            | 4.72               | 3.50                 |
> | Gemini 2.0 Flash    | 30.22                 | 45.49               | 17.5                          | 9.45               | 5.00                 |
> | Gemini 2.5 Pro      | **66.37**             | **69.94**           | **35.35**                     | 10.16         | 1.50                 |
> | Gemini 2.5 Flash    | 29.16                 | 52.36               | 26.77                         | 2.38               | **4.0**              |
>
> **Long Audio (Higher is better for all tasks)**
>
> | Model               | Word Localization (F1) | NE Localization (F1) | Advertisement Localization | Speaker Num. Estimation (Acc) | Event duration (Acc) | Entire Duration (Acc) | Emotion ranking (Acc) | Emotion reasoning (Acc) |
> |--------------------|-----------------------|---------------------|----------------------------|-------------------------------|--------------------|----------------------|----------------------|------------------------|
> | Gemini 2.0 Flash    | 1.12                  | 2.97                | 4.93                       | 8.00                           | 1.49               | 0.50                 | 26.28                | 54.54                  |
> | Gemini 2.0 Pro      | 0.19                  | 2.14                | 0.15                       | 8.5                           | 1.49               | 0.00                 | **32.00**                | 64.29                  |
> | Gemini 2.5 Pro      | **15.79**             | **11**              | **11.48**                  | **35.18**                     | 1.98               | 1.5 0                 | 25.00                | **66**                 |
> | Gemini 2.5 Flash    | 6.16                  | 1.93                | 17.8  | 14.07                         | **2.48**           | 0.5                  | 19.2                 | 56.8                   |

---

> ### Author Response · Authors · 2025-11-21
>
> **Word timestamps post-correction not sufficient**
> - Forced aligners could be prone to errors, but in our case, these were rare and straightforward to correct. Our approach is consistent with prior work that used similar methods for annotating NE timestamps on short audio tasks (SLUE Phase-2: https://aclanthology.org/2023.acl-long.496.pdf), (Dynamic-SUPERB Phase-2: https://openreview.net/forum?id=s7lzZpAW7T). These studies support our finding that forced-aligner–derived timestamps can provide sufficiently high-quality ground truth for our purposes.
>
> **There's almost no coverage of non-speech environmental or musical structure.**
> - We agree that a broader range of audio domains would be valuable. Our focus was on realistic, understandable tasks we could reliably evaluate using long-form inputs. It was not feasible to directly include long-form music or full-length entertainment media due to licensing and data availability constraints. However, we note that these domains are still indirectly represented in our data sources. For example, several samples come from movie-review videos that include conversational speech, background sounds, and brief music segments. While BLAB does not explicitly cover isolated music or entertainment content, it does capture mixed audio conditions that reflect aspects of these domains as suggested.
>
> **Human Annotation Details**
> All tasks were fully human-audited. For every task, all items undergo complete human verification, except word localization, where a small sample is used to assess error rates and error types, which were minimal.
> For both NE and Advertisement localization tasks, we start with a cascaded approach to get initial ground truth annotations. Then we perform human verification to correct any errors, removing falsely labeled segments, adjusting timestamps as needed, and adding any missing segments.
> For speaker number estimation, each audio file is labeled by two annotators, and a third reviews both annotations and selects a final count label. Because the audio is long and complex to annotate, the speaker counts varied, and disagreements were common. In the majority of cases, annotators argued that multiple counts could be valid, so we retained a range of count labels and adjusted our evaluation metric. For the event duration task, one annotator first identifies candidate events in the audio, comes up with questions, and estimates their durations. The emotion task follows a similar two-step procedure: one annotator extracts emotionally salient segments and drafts questions, and another reviewer validates them. This setup is consistent with the data curation for existing audio benchmarks like MMAU (https://arxiv.org/pdf/2410.19168) and MMAR (https://arxiv.org/abs/2505.13032), where an annotators come up with questions, and there is a final round of review of both questions and labels.
> We will revise the relevant sections in the paper draft to make this clearer.
>
> **NE & Ad localization are derived by running text-only LMs on transcripts, then mapping spans back via WhisperX timestamps. This makes many items solvable from text alone and entangles evaluation quality with ASR/FA errors rather than acoustic understanding.**
>
> - It is correct that our current data curation pipeline for NE and Advertisement localization uses text-only LMs on transcripts and maps the predicted segments back to forced aligner timestamps. To increase evaluation quality, we manually validated the derived segments and respective timestamps and corrected cases where the FA or model introduced errors.
> - Regarding the concern that these tasks could be solvable by text alone,  BLAB is designed to evaluate large audio language models end-to-end, testing true multimodal and temporal understanding without relying on external tools having prior access to transcripts. For advertisement localization, acoustic cues such as jingles, background sounds, or speaker emphasis can guide the model to make more accurate predictions. Besides, other tasks in the benchmark that are temporal, speaker, and emotion-related, already test for acoustic understanding and reasoning.

---

### Meta-Review · Area_Chair_rFCp · 2026-01-06

**Summary:**

This paper proposes, to the best of my knowledge, the first benchmark specifically designed for evaluating long-audio understanding in audio LLMs. The benchmark includes multiple tasks intended to highlight the importance and challenges of long-duration audio processing.
However, several key concerns raised by the reviewers remain insufficiently addressed:


Evaluation protocol for short-context models
Reviewers t4yw-W1 & W2, NZWq-W1 & W2, and 1ctL-W1 note that the authors have added evaluations of several recent models (e.g., Gemini 2.5, GPT-4o, and AF3). From the reported results, it appears that models with limited audio context length are evaluated without adopting a sliding-window strategy. While sliding-window inference is indeed unsuitable for certain global tasks such as speaker number estimation, it should be applicable to several other benchmark tasks, including word localization, named-entity localization, event duration estimation, and total duration estimation. As such, a comparison against sliding-window baselines would be necessary for a fairer assessment.


Reliability of transcriptions and temporal annotations
Reviewers t4yw-W3 & W4 raise concerns that the word-level transcriptions and their corresponding timestamps may contain substantial errors and are not carefully verified or corrected by human annotators. Although the authors argue that similar annotation standards were used in MMAU and MMAR, such errors are likely to have a more pronounced impact in long-audio scenarios, where error accumulation and temporal drift can significantly affect downstream task performance. This issue remains insufficiently analyzed or mitigated.


Overall, while the benchmark is timely and potentially impactful, these unresolved issues limit the strength of the conclusions drawn from the current experimental results.

**Reviewer Concerns:**

The concerns raised by Reviewer t4yw-W1 & W2, Reviewer NZWq-W1 & W2, Reviewer 1ctL-W1, and Reviewer t4yw-W3 & W4 remain unresolved, whereas the remaining concerns have either been fully addressed or are not applicable (e.g., Reviewer 1ctL-Q1 regarding the comparison between CHiME and BALB).

**Reviewer Scores:**

I expect that all reviewers will maintain their original ratings, as the rebuttals did not adequately resolve the major concerns.

---

### Decision · Program_Chairs · 2026-01-26

Reject